# Generating Artificial Reverberation via Genetic Algorithms for Real-Time Applications [note 1]

**DOI:** 10.3390/e22111309

**Published:** 2020-11-17

**Authors:** Edward Ly, Julián Villegas

**Affiliations:** Computer Arts Laboratory, University of Aizu, Aizu-Wakamatsu 965-0006, Japan; julian@u-aizu.ac.jp

**Keywords:** convolution reverb, genetic algorithms, impulse responses, room acoustics, signal processing

## Abstract

We introduce a Virtual Studio Technology (VST) 2 audio effect plugin that performs convolution reverb using synthetic Room Impulse Responses (RIRs) generated via a Genetic Algorithm (GA). The parameters of the plugin include some of those defined under the ISO 3382-1 standard (e.g., reverberation time, early decay time, and clarity), which are used to determine the fitness values of potential RIRs so that the user has some control over the shape of the resulting RIRs. In the GA, these RIRs are initially generated via a custom Gaussian noise method, and then evolve via truncation selection, random weighted average crossover, and mutation via Gaussian multiplication in order to produce RIRs that resemble real-world, recorded ones. Binaural Room Impulse Responses (BRIRs) can also be generated by assigning two different RIRs to the left and right stereo channels. With the proposed audio effect, new RIRs that represent virtual rooms, some of which may even be impossible to replicate in the physical world, can be generated and stored. Objective evaluation of the GA shows that contradictory combinations of parameter values will produce RIRs with low fitness. Additionally, through subjective evaluation, it was determined that RIRs generated by the GA were still perceptually distinguishable from similar real-world RIRs, but the perceptual differences were reduced when longer execution times were used for generating the RIRs or the unprocessed audio signals were comprised of only speech.

## 1. Introduction

In most situations, sound is perceived within enclosures such as rooms, large or small venues, caves, etc. These enclosures imprint their characteristics (i.e., their transfer functions) on the sound depending on where the sound source and the listener are located within the enclosure. The direct sound travels the shortest path between the source and the listener, while other paths may include reflections off the boundaries of the enclosure. Hence, reflected sounds arrive later than the direct sound and are, in general, weaker because of the energy absorption of the reflecting surfaces as well as the transmission medium. The collection of early and late reflections is usually referred to as “reverberation”. While this phenomenon is often detrimental to speech intelligibility [1], it is frequently desired in music production as an expressive tool [2].

The most direct form of including reverberation in audio recordings is by capturing the sound directly in spaces with the desired acoustic characteristics. Alternatively, one can record the room’s characteristics, or its Impulse Response (IR), with which a dry audio signal (one that is anechoic or contains a very short reverberation) can be convolved. However, recording (even just the IRs) in a majority of venues can be expensive and may not always be possible, so other methods to add artificial reverberation have been devised. These methods yield reverberations similar to those captured in venues without apparent quality detriment (in the next section, a few of these techniques will be described in more detail). Regardless of the alternatives for adding reverberation, uniqueness is a characteristic often sought for artistic purposes. However, creating new and unique IRs with desired room characteristics can be a difficult task without correct or adequate methods. The purpose of this research is to introduce a new method for creating IRs using Genetic Algorithms (GAs). This method should produce unique results each time while satisfying some constraints imposed by a user. We also investigate the suitability of this approach by measuring the elapsed time, fitness, and subjective accuracy of the IRs thus generated.

Parts of this article were previously published in [3]. The core of the research remains the same, but has since been expanded upon for this article with the following changes: additional background information, updates to the methodology and subsequent experimental results, an additional (subjective) evaluation method, an expanded conclusion, and general rewording for improved clarity. Additionally, from the original submission, we have improved our solution by adding binary control over the Interaural Level Difference (ILD), refining the computation of clarity ($C_{80}$), and adding predelay as a parameter to the generated IRs.

The rest of this article is organized as follows: Section 2 provides some background information and related work regarding convolution reverb, impulse responses, and the usage of evolutionary algorithms in computer music research. Section 3 delves into the GA implemented to generate artificial IRs and the plugin technology used to embed the GA within a real-time audio effect. Section 4 presents objective and subjective evaluations of our solution. Lastly, Section 5 and Section 6 provide interpretations of these results followed by concluding remarks.

## 2. Background

When producing reverberation artificially, it is often difficult to reproduce some acoustic characteristics of physical spaces. Even when the IR of a room is known, synthesizing a similar IR is challenging. In this section, we will explain how artificial reverberation can be applied to an audio signal in real-time using IRs or other techniques, as well as what a GA is and how it works. Then, we will provide a brief overview of previous research in evolutionary computing for musical applications, as well as recent advances in digitally simulating reverberation via convolution with IRs.

### 2.1. Reverberation Techniques

Some of the most common techniques for digitally simulating reverberation involve the use of delay lines, comb filters, feedback delay networks, etc. Fundamentally, all of these Digital Signal Processing (DSP) devices work similarly. A simple delay-line-based reverberation modifies an audio signal by adding a delayed version of the same signal to itself. The delayed version is also attenuated, simulating a single sound reflection. This delay can be repeated many times to construct a synthetic IR, but having to program every reflection can be very inefficient. Comb filters, in feedforward form, work the same way, but a feedback component can be included as well. In this case, the output signal can be recursively fed back into the comb filter as an input signal. Given adequate parameters, a comb filter can produce an infinite number of decaying reflections, or a series of echoes, at regular time intervals. The first known implementation of reverberation in software was developed in 1961 by Schroeder, who used a series of recursive comb filters in parallel to simulate wall reflections with exponential decay [4]. Depending on the number of delay objects, however, the resulting IR can be difficult to modify, as the parameters of each delay object only control either a single reflection or a series of related reflections. This makes the process of modifying the IR to match certain room characteristics cumbersome at best.

Convolution reverb works differently in that instead of representing the IR of the desired room as a series of delay objects, the IR information is stored directly as audio, an array of samples in the time domain, with each sample value representing the gain and polarity of the sound pressure signal after some delay. Then, assuming that an audio signal and the IR are sampled at the same rate, the two signals can be combined (via the convolution operation) to produce a reverberated signal. In order for this process to be implemented in real-time, however, two modifications to the convolution operation have been proposed by Zölzer [4]: First, the convolution operation is replaced by frequency-domain filtering (also called “fast convolution”). In this operation, an audio signal and the IR are transformed from the time domain into the frequency domain (via Fast Fourier Transform—FFT), the two transformed signals are multiplied element-wise, and the result is transformed back into the time domain (via the inverse FFT) to produce the output signal. Second, “partitioned convolution” is used [5]. This is an operation in which the audio signal and the IR are first partitioned into smaller blocks before fast convolution is performed on each block individually. Each of the reverberated blocks are then recombined to construct the output signal.

Regardless of the specific algorithms used, current solutions (often software plugins) that implement convolution reverb are limited by their selection of pre-loaded IRs or enclosures that a user can choose from. While the ability to modify the existing IRs may be offered by some plugins, a plugin in which new IRs can be generated from scratch seems to be yet to be discovered. That is the void that the “Genetic Reverb” plugin presented in this research is seeking to fill.

### 2.2. Genetic Algorithms

Within Artificial Intelligence (AI), there is a sub-field known as evolutionary computation, which consists of a family of algorithms inspired by Darwin’s theory of evolution [6] for global optimization problems. Genetic Algorithms [7] are one of the simplest and and most well-known cases of these algorithms; they are multi-point search algorithms where potential solutions are pooled together into a “population”, with each solution representing an “individual”. Each solution is often encoded as a “genome” consisting of one or more “genes”, which are modified by the GA to create new solutions. The GA itself consists of a single execution loop comprising three operations: selection, crossover, and mutation. Each pass through the loop, or “generation”, modifies the population in such a way that, over time, the average “fitness” of the population often increases.

At the start of a generation, each individual is given a numerical “fitness value” to serve as a metric of its “goodness” relative to the other individuals. With these values, the best-fit individuals can be determined and kept in the population, while the least-fit individuals can be removed. Determining which individuals are kept is also known as the selection operation. In the crossover operation, the removed individuals are replaced by “offspring,” new individuals that are derived by combining the genomes of two random individuals in the remaining population. Finally, in the mutation operation, each gene has a very small probability (usually <1%) of having its value changed entirely. Such an operation is necessary to maintain some amount of diversity within the population. Otherwise, the population could tend towards a solution whose fitness value is merely a local maximum when other solutions with better fit may exist elsewhere in the solution space.

While such a loop can run indefinitely, the GA must eventually stop and return an output, so one or more terminating conditions are usually implemented. The two most common terminating conditions are whether a predetermined number of generations has passed or whether the best-fit individual is “good enough” within the context of a problem. When a terminating condition has been met, the solution with the best fit is returned by the GA.

#### 2.2.1. Genetic Algorithms in Sound and Audio

The idea of applying evolutive techniques to real-time DSP is not new (see, for example, [8]), but compared to other techniques in artificial intelligence, there seems to be little research and few actual implementations. The implementations that exist are not just limited to using GAs, either. Genetic Programming (GP) [9], for instance, is a technique similar to GAs, except that entire graph structures (e.g., a DSP tree) are subjected to evolution. One example comes from Macret and Pasquier [10], who generated sound synthesizers in Pure-Data (Pd) via Mixed-Typed Cartesian Genetic Programming (MT-CGP). Pure-Data is a visual programming language mostly for audio DSP, with a graphical interface that visualizes each program as a directed graph structure [11]. Their approach is useful for generating new sounds and instruments that closely resemble other target sounds, but the average end user may not be familiar with Mel-Frequency Cepstral Coefficients (MFCCs), which are the features used in their Graphical User Interface (GUI) as input parameters for the MT-CGP to represent the target sound.

Collins [12] goes further by providing a software library built in SuperCollider [13] that can apply GAs to both sound synthesis and real-time DSP, including filtering and reverberation. Using SuperCollider’s built-in GUI and programming tools, a framework for other developers to build software applications using GAs was provided. Collins does not mention any ability for a GA to evolve a population of IRs for convolution reverberation, but rather other devices for reverberation techniques, such as multi-tap delay lines, feedback delay networks, and Schroeder reverberators (which can be described as a series of all-pass and comb filters). In addition, the SuperCollider environment requires the use of *scsynth*, a dedicated audio server for real-time DSP computation. As of November 2020, a *SuperColliderAU* wrapper is available to embed *scsynth* into an Audio Unit (AU) plugin, but this technology is only available in the MacOS operating system. A similar wrapper to embed *scsynth* into a Virtual Studio Technology (VST) plugin (a popular cross-platform audio software interface widely used in music production) is not yet available.

Additionally, a Pd program for sound localization and spatialization using GAs has been proposed by Fornari et al. [14]. With their GA, they first calculate the Interaural Time Difference (ITD) given sound intensity and azimuth angle as input parameters. Then, with this ITD, a Sonic Localization Field (SLF) with which sound can be perceived at a particular location in space can be generated.

DSP applications for sound spatialization are, in general, limited because a free-field is assumed; i.e., no reverberation is considered. In many cases, an audio signal is directly convolved with a Head-Related Impulse Response (HRIR) that captures only the transformations caused by the upper body of a listener to a sound in the absence of an enclosure (e.g., [15]). However, these applications can benefit from artificial reverberation, since, when the direct sound is combined with reverberation, or at least with late reflections (the tail of the reverberation), the externalization of audio sources seems to improve [16].

#### 2.2.2. Simulating The Impulse Response of a Box-Shaped Room

An image method for simulating the IR of a box-shaped room was originally proposed by Allen and Berkley [17]. In their method, such a room was projected onto three-dimensional space to determine which sound reflections contributed to an IR. The same projection was also used to calculate the effective distance and associated gain for each contributing reflection. As a metaphor, one can imagine a listener and a sound source located arbitrarily within a room with mirrors on all sides. Such mirrors would create an infinite number of images of the source outside the initial boundaries of the room, and the listener could see the source reflected an infinite number of times. Then, each source image would contribute a single sound reflection, and the sound from each object image would then be delayed by some time before it reaches the listener. Such a delay can simply be calculated as the perceived distance between the listener and the image divided by the speed of sound. Additionally, each reflection will lose power depending on the number of “walls” the sound has to “pass through” before reaching the listener’s location. Practically, reflections greater than a certain order are ignored in software implementations, as such reflections become too quiet to be perceived or numerically represented.

Many extensions and applications based on the Allen and Berkley method would later be proposed. One such implementation is provided by Habets [18] in Matlab [19]. More recent revisions to the image method that tackled these performance issues have also been proposed, such as those by Kristiansen et al. [20] and McGovern [21]. Kristiansen et al. use the image method to extend the length of an existing IR by using known lower-level reflections to calculate higher-level ones, while McGovern [21] improves on the original image method algorithm by using look-up tables and sorting to prevent unnecessary calculations, greatly reducing computation time.

Initially, the fast image method proposed by McGovern was implemented as the baseline model for generating an initial population of IRs in the “Genetic Reverb” plugin. However, the GA ultimately outputs IRs that no longer represent box-shaped rooms regardless of the nature of the initial population, so there was no reason why other (non-box shaped) models could not be used in the first place either. Thus, a new method for generating IRs modeled after recorded IRs was devised for our solution. The following section describes the method used in our GA in detail.

## 3. Method

Typical GAs often evolve a population of individuals that can be encoded as a series of binary or real-numbered values. As audio is also stored as an array of real values, it seems natural to attempt to evolve IRs or even entire audio signals as well. While it is theoretically possible to generate IRs using other evolutionary algorithms similar to GAs, we ultimately chose GAs for the sake of simplicity and familiarity with this particular type of algorithm. This section details each step of our GA process for generating an IR as well as some notes regarding the implementation of our GA in software.

### 3.1. Genetic Algorithm

A classic GA was implemented on a population of room IRs, where the genome for each IR individual consists of an array of real-valued genes in the range of [−1,1] in the time domain. The first gene at the start of the IR is also assumed to be non-zero and to represent the first reflection. Multiple applications of the genetic operations will then change the amplitude of existing reflections until a created IR closely adheres to the user constraints. Figure 1 provides an overview of the GA process as it pertains to our solution.

Five acoustic parameters that a user can impose on an IR in our solution are summarized in Table 1. The first three parameters ($T_{60}$, Early Decay Time (EDT), $C_{80}$) are defined under the ISO 3382-1 standard [22], while the other two parameters, Bass Ratio (BR) and Initial Time Delay Gap (ITDG), are some of many additional parameters and attributes proposed by Beranek [23]. These parameters not only enable control over the overall shape of the IR, but are also parameters that a typical end user may understand and control.

The $T_{60}$ parameter is perhaps the most important, since it specifies the overall length of the reverberation (i.e., how long the lingering sound is heard). In general, a long $T_{60}$ correlates with large enclosures or enclosures with walls featuring high acoustic reflection coefficients. The Initial Time Delay Gap (ITDG) is a similar measure for determining the “intimacy” of an enclosure, with longer ITDGs corresponding to wider or taller enclosures [23] (pp. 513–516). Meanwhile, the EDT characterizes the initial rate of decay of the IR, and it is generally considered more subjectively relevant than $T_{60}$, since the early reflections are more salient to the perception of reverberation than the late reflections [22].

Clarity ($C_{80}$) measures the relative “intelligibility” of the original audio source within its surrounding reverberations. High clarity indicates how “clearly” or “intelligible” a sound source can be heard. Generally, IRs with long $T_{60}$ tend to have low $C_{80}$ values. Finally, the BR is a measure of the frequency content of an IR. Is is defined as the ratio of the reverberation times (RT) within four consecutive octave bands, with center frequencies at 125 Hz, 250 Hz, 500 Hz, and 1 kHz [23] (p. 512):
(1)$$BR=\frac{RT_{125\mathrm{Hz}}+RT_{250\mathrm{Hz}}}{RT_{500\mathrm{Hz}}+RT_{1\mathrm{kHz}}}.$$

High BR values indicate that low frequencies are more prominent in the reverberated sound compared to high frequencies.

#### 3.1.1. Initialization

In the initialization step, a population of IRs is generated using a custom Gaussian noise method based on the scheme proposed by Zölzer [4] (p. 144). This noise is typically present in observed IR recordings (mainly from the signal-to-noise ratio of the recording apparatus). The IR population is stored as a two-dimensional matrix, with each column representing an individual IR, and the number of columns is equal to the desired size of the IR population. The length of each column (i.e., the number of samples in the IR) is determined by the reverberation time $T_{60}$ of the desired IR along with the sampling rate of the IR to generate. In our solution, the sampling rate was set to a constant 16 kHz in order to minimize computation time.

Within each IR, each sample is initialized as a random number from the standard normal distribution. Then, each IR is manipulated in such a way in an attempt to produce IRs that resemble actual IRs that have been recorded in the real world. This is done through the following steps:
Reduce the gain of the entire signal by a constant amount (a random factor between 0.2 and 0.7). This is to introduce some variation in regards to how much of the sound is absorbed by the enclosure.Change the value of certain samples to be closer to ±1. The probability that a sample is chosen at time *t* (in s) is
(2)$$P\left(t\right)=1-p^{t}$$
for some randomly chosen probability constant p∈(0,1), simulating the higher density of late reflections compared to that of early reflections. In addition, the new values of these samples are randomly generated from a normal distribution (μ=1,σ=0.05), along with a 50% probability of each of these values being positive or negative. This is to emphasize the presence of the room reflections over the diffused sound.Apply exponential decay to the gain of the entire IR (the rate of which is inversely proportional to the input $T_{60}$ value), simulating the absorption of sound from the surrounding walls.Apply a second-order Butterworth band-pass filter, where the cutoff frequencies of this filter are randomly chosen (31.25–500 Hz for the lower and 0.5–8 kHz for the upper bound). Such a filter is normally applied to recorded IRs as well [24].Additionally, low-pass Gaussian noise (cutoff frequency at 250 Hz) with 12 dB/octave rolloff is applied. This spectral tilt was observed in the tail of real IR reverberations [24], and it is perceptually relevant [25].

The steps and values specified above were arbitrarily chosen, and at least impressionistically, these values produced IRs that best resemble IRs that have been recorded in the real world, such as those in the OpenAIR database [24]. At the end of this process, an initial population of IRs is seeded and used as a starting point from which other parameters can be sought.

#### 3.1.2. Fitness

Each IR in a population is then assigned a fitness value based on how closely it matches user’s specifications. In our implementation, an “error” or “loss” value is assigned, with figures toward zero representing a better “fit,” since a value measuring the differences between IRs is desired. First, we calculate the *z*-score for each of the standard room acoustic parameters summarized in Table 1 (except for ITDG, since it can be generated with exact accuracy) using the desired IR parameter values as the mean for each *z*-score. The error value of each IR is then determined by the sum of the absolute values of all the *z*-scores (i.e., the total number of standard deviations away from the optimal solution). Because of the differences in units and in numerical scale between IR parameters, this error value allows one to weigh the individual parameters equally and also detect and remove outlier IRs easily.

As for determining the acoustic parameter values for each IR, it is possible to compute them directly and then compare them with the desired values to find the closest match. For instance, $T_{60}$ can be calculated using the IR’s Schroeder curve S(t) (also known as the energy decay curve) [26], formally defined as the integral of the square of an IR h(τ) from time τ=t:
(3)$$S\left(t\right)=\int_{t}^{\infty }h^{2}\left(\tau \right)\;d\tau .$$

By expressing S(t) in dB, it is possible to locate the instances where the sound energy decays 5 dB and 35 dB from the initial level. This time difference is known as $T_{30}$, and doubling it yields an accurate approximation of $T_{60}$. This is a common method of calculating $T_{60}$, as the signal-to-noise ratio of a measured IR is often less than 60 dB, and samples within the noise floor should be avoided [22].

EDT can be computed in a similar manner, except that the time difference is estimated between the arrival of the direct sound (0 dB) and the time when the total energy decays 10 dB. Clarity ($C_{80}$) can be determined using Schroeder curves as well. $C_{80}$ is defined in terms of S(t) as
(4)$$C_{80}=10\log_{10}\left(\frac{S(0)-S(0.08)}{S(0.08)}\right)\left[\mathrm{dB}\right],$$
where t=0 refers to the time of arrival of the direct sound. For BR, instead of calculating the reverberation times of the relevant octave frequency bands, the IR is transformed into the frequency domain via the Discrete Fourier Transform (DFT) before the sound energy ratio (in dB) between the 125–500 Hz and 0.5–2.0 kHz bands is computed. While the actual values between the two methods may differ, the implemented algorithm is faster to execute with no apparent detriment in accuracy. In addition, expressing the Bass Ratio in dB (as opposed to a unit-less value) may help naive users to better understand this parameter.

#### 3.1.3. Genetic Operations

In each generation, the IR population undergoes three successive operations (i.e., selection, crossover, and mutation) in order to search for a better match. For our solution, truncation selection, random weighted average crossover, and Gaussian multiplication for mutation have been chosen for the GA. Other alternatives include: rank selection, tournament selection, *n*-point crossover, and permutation mutation (for a more comprehensive list of genetic operations, the interested reader is referred to [27]). These operations were chosen based on maintaining both computational efficiency as well as acoustic variety in the IR population. Additionally, some methods were simply not viable within the context of creating artificial IRs. One-point crossover, for example, would create a child IR where the size of the virtual room changes after some time, which, in most cases, would not be desirable in a plausible IR.

Truncation selection is one of the simplest selection methods: The bottom percentile of IR individuals (those with the worst fit) are removed from the pool each generation. For our solution, a selection rate of 0.4 was chosen, meaning that individuals with error values below the 60th percentile are removed from the population. The removed IRs are then replaced with new ones through a random weighted average crossover operation: Given two random parent IRs $h_{1}\left(t\right)$ and $h_{2}\left(t\right)$, and a random weight factor w∈(0,1), the offspring IR $h_{1,2}\left(t\right)$ is defined as
(5)$$h_{1,2}\left(t\right)=wh_{1}\left(t\right)+\left(1-w\right)h_{2}\left(t\right).$$

This operation effectively blends the two parent rooms together to create a new room with its own acoustic parameter values from which a fitness value can be determined. Finally, in the mutation operation, each sample has a very small probability (1 out of every 1000 samples) of changing its value completely. To mutate the IRs in particular, a 50% probability of increasing or decreasing in magnitude as well as a 50% probability of switching sign for each sample to mutate is desired. To achieve this, mutation is performed via Gaussian multiplication, where each sample is multiplied by a normally distributed random number (μ=0,σ≈1.483).

#### 3.1.4. Termination

After each round of genetic operations, the fitness of the new IR population is recalculated, but this is when the GA determines whether or not it should continue modifying the population or stop and return the best-fit IR found so far. This decision is based on three terminating conditions:
The fitness value of the best IR found is below a certain threshold value,The fitness value of the best IR found does not decrease after a certain number of generations (specified by a “plateau length” parameter) has passed, orA predetermined limit on the total number of generations to execute has been reached.

The fitness threshold, the plateau length, and the maximum number of generations, therefore, are all parameters within the GA intended to limit computation time. Since an end user might not understand what a GA does, however, including these parameters directly in the plugin’s GUI might not be user-friendly. For this reason, a “Quality” setting was used to abstract these parameters away while still providing the user some control over the computation time of the GA and, hence, the fitness values of the IRs. Table 2 lists the current mapping between the “Quality” parameter and the GA parameters.

### 3.2. Implementation

Genetic Reverb was implemented as a subclass of Matlab’s built-in audioPlugin and System classes. These classes can be compiled into a VST 2 plugin compatible with both Windows and macOS operating systems. Table 3 presents a list of all parameters that can be controlled by a user, their valid ranges, and a brief description of each.

For stereo signals, two separate IRs are always active and assigned the left and right channels for convolution. Depending on whether the plugin is in “mono” or “stereo” mode, however, one or two new IRs can be generated by the GA, which is triggered each time the “Generate Room” button is pressed. In mono mode, the single IR is copied for both channels to use. In stereo mode, the GA is executed twice, each producing a different IR for the left or right channel. Due to the random nature of the GA, the Root Mean Square (RMS) levels of the two IRs may differ. We can limit this difference to be less than the maximum ILD, approximately 20 dB [28] (p. 230), to preserve the perception of sound localization. When the perceived loudness of each IR must be roughly the same, users have the option of equalizing the RMS levels of the IRs.

Then, we reinstantiate the predelay to the IRs, since a typical IR has some delay before the first reflection reaches the ear. While this predelay was previously used to determine the $C_{80}$ value of the IRs, the delay itself is not added until after the IRs are generated by the GA. Finally, the new IRs are resampled from 16 kHz to the sampling rate of the host application before convolution with the input signal, since the sample rates of the IRs and the input signal must match.

For real-time convolution reverb, an object that performs partitioned fast convolution is readily available in Matlab [19]. In the partitioned convolution, the IR is broken up into blocks of 1024 samples each, which are convolved with the audio signal separately and then added back together (via the “overlap-save” method) to produce the output signal. This reduces the latency of the convolution from the entire length of the IR to approximately 1024/f seconds, where *f* is the sample rate of the audio signal. One limitation of this object is that the length of the array containing the filter coefficients (i.e., the number of samples in the IR) cannot be changed dynamically. As a workaround, we created multiple copies of this object, each with different lengths for the filter arrays. Then, depending on the current value of the $T_{60}$ parameter, the object with the most appropriate length is chosen. Finally, the newly generated IRs can be saved into that specific object, which can then be set as active for use with partitioned convolution with the input signal. Figure 2 illustrates the basic flow of data and audio within the plugin, while Figure 3 shows its current user interface. Our implementation along with some demonstrations are available from the Appendix A.

## 4. Evaluation

While we were successful in implementing a GA that can produce synthetic IRs, this alone does not tell us whether the GA can produce IRs that actually adhere to user constraints. In other words, how closely does the output IR match what the user expects in terms of the resulting reverberation? To that end, we performed one objective and one subjective evaluation. For the sake of simplicity, only monophonic IRs were considered in these tests.

### 4.1. Objective Evaluation

We were interested in finding how close to zero the loss values could reach for every IR returned by the GA given certain acoustic parameters. We were also interested in the elapsed time to reach those loss values. These tests were run locally on a Windows 10 Home laptop computer equipped with an Intel Core i9-8950HK six-core notebook CPU running Matlab version R2020b. In these evaluations, Matlab’s built-in time-keeping tools (i.e., the tic/toc functions) were used to record the execution times of the GA.

In a first test, the values for EDT, $C_{80}$, BR, and ITDG were randomly selected within the valid range of the plugin for each IR, while $T_{60}$ was restricted to t∈{0.625,1.25,2.5,5} s in order to measure the relationship between $T_{60}$ and computation time. This process was repeated four times, one for each of the four possible “Quality” settings, with a population of 250 IRs being generated for each combination of $T_{60}$ and “Quality” setting.

In a second test, the acoustic parameter values were limited to those from real IRs to determine how well the artificial IRs from the plugin could emulate real IRs. For this, an IR was arbitrarily chosen from the OpenAIR database [24], and its acoustic parameter values were calculated using the same algorithms used in the GA. The selected IR was recorded inside the York Guildhall Council Chamber (source position 1, receiver position 1, channel 1), and its acoustic values are $T_{60}=0.884$ s, EDT =0.133 s, $C_{80}=4.678$ dB, BR =−1.233 dB, and ITDG =5.2 ms. Then, 250 IRs were generated for each of the four “quality” settings, adjusting the acoustic parameters to the above values.

#### Results

Figure 4 provides a summary of the output IRs from the first test, reporting the distributions of loss values and elapsed times, respectively, for each quality setting and $T_{60}$ decay time via violin plots. Like box plots, violin plots include a center white dot and surrounding gray bar to indicate the mean and interquartile range (IQR), respectively. Violin plots, however, are more informative than box plots with the inclusion of probability density functions that are smoothed via kernel density estimation [29]. Figure 5 provides a similar summary for the output IRs from the second test.

### 4.2. Subjective Evaluation

For the subjective evaluation, the goal was to determine whether or not there were perceptual differences between audio signals convolved with real-world IRs and artificial IRs generated with the GA. For this evaluation, three IRs from the OpenAIR database with different $T_{60}$ decay times were chosen. For each one, its acoustic parameter values were calculated, and then two artificial IRs were generated based on these values. One IR was generated using the “High” quality setting, while the other was generated using the “Max” quality setting. These IRs were then convolved with two mono audio signals: one from a male speaker at 16 kHz, and one from a dry riff of a synthesized drum kit at 48 kHz, to produce 12 unique reverberated audio stimuli. The IRs were generated at a default sample rate of 44.1 kHz, and then resampled to match the sample rate of the audio signals.

With these audio samples, an ABX test was created. An ABX test consists of a series of questions where two similar but different audio samples labeled *A* and *B* are presented, along with a third audio sample labeled *X* [30]. The audio sample labeled *X* is identical to either sample *A* or sample *B* at random, and participants were asked to determine whether *X* was equal to *A* or *B*. Each possible real IR and artificial IR pair combination was presented six times in total, three with the speech stimuli and three with the drum kit stimuli, with random labeling of *A*, *B*, and *X* for each of the 36 total questions. The survey was split into two blocks, with the speech samples being presented in the first block before the drum kit samples in the second block. The order of the questions within each block was randomized for each participant. Participants were asked to conduct the experiment in an environment free of external noise or distractions, and to refrain from creating noise such as eating or chewing gum. However, since the survey was conducted online (as a COVID-19 precaution), such conditions cannot be guaranteed.

After the conclusion of the survey period, 26 participants (21 male and 5 female) completed the survey, while five more volunteers started the survey, but did not complete it. The latter were removed from the statistical analyses. Participants who completed the survey were financially compensated for their collaboration. Before the start of the experiment, participants were asked about their age, sex, and amount of musical experience. The ages of the participants who completed the survey ranged from 20 to 41, although 24 of the 26 participants were between 20 and 25 years old. Seven of the participants declared that they had no musical experience, ten responded with 1–3 years of experience, and the remaining nine declared four years of experience or more. Permission for performing this experiment was obtained following the University of Aizu ethics procedure.

#### Results

A script was developed in R [31] to determine whether or not variables such as age or quality settings had a significant effect on the rate of correct answers in the experiment. A Generalized Linear Mixed Model (GLMM) was implemented to predict such effects for the binomially distributed data. Although there are other alternatives to analyze this kind of data, it has been shown that GLMMs are suitable for these analyses [32]. For this experiment, age, sex, years of musical experience (‘exp’), the repetition order of a question (‘rep’), and subjects (‘id’) were considered random effects. Meanwhile, the ‘program’ (speech vs. drum riff), IR ‘quality’ (High vs. Max), and whether sample *X* was created from a real or artificial IR (‘x.type’) were deemed fixed effects. After multiple pairwise comparisons of various models via Analysis of Variance (ANOVA), it was determined that only ‘program’ and ‘quality’ had a significant effect on the subjective responses. The $\chi^{2}$ values, degrees of freedom (DF), and *p*-values obtained through such comparisons are summarized in Table 4.

A post-hoc analysis based on Tukey’s Honest Significant Difference (HSD) test comparing the Estimated Marginal Means (EMMs) of each level of the interaction was conducted with the emmeans library [33], with the degrees of freedom being asymptotically computed for this analysis. Tukey’s test showed that, regardless of quality setting, the rate of correct answers for the drum riff was significantly higher than that of speech (z=5.926,p<0.001). Additionally, regardless of “program,” the rate of correct answers given “High” quality IRs was significantly higher than that of “Max” quality IRs (z=2.414,p=0.016). These findings are summarized in Figure 6a, which shows the mean and 95% confidence interval of the probability of a correct answer by quality setting and program. We can also conclude from Figure 6a that the subjective responses were significantly different from random, since the rate of correct responses is significantly higher than 50% in all cases.

A further inspection of the responses revealed that twelve participants returned perfect or near-perfect scores (those who answered 100% correctly for at least three out of four question groups, where each question is classified based on the program and the quality setting of the artificial IR). We ran a similar statistical analysis without these participants (or “super-classifiers”) to verify that the previous results still hold for the remaining participants.

Despite a decrease due to the removal of super-classifiers, the rate of correct responses is still significantly higher than 50% in all cases. Furthermore, similar GLMMs with the reduced dataset still show that the simplest statistically significant prediction model remains the same as that shown previously. The $\chi^{2}$ values, degrees of freedom, and *p*-values obtained through similar pairwise comparisons of the GLMMs are summarized in Table 5.

Tukey’s HSD test also yielded similar results, with the rate of correct answers for the drum riffs being higher than that of speech (z=5.332,p<0.001, across levels of “quality”). Similarly, the rate of correct answers given “High” quality IRs was still significantly higher than that of “Max” quality IRs (z=2.035, p=0.042, across levels of “program”). These findings are summarized in Figure 6b.

## 5. Discussion

In short, our solution is not able to produce an adequate IR 100% of the time. However, this outcome was to be expected for reasons that will be explained below. Both the objective and subjective evaluations also provide some further insight into specific areas where the GA may be lacking, along with possible solutions.

### 5.1. Objective Evaluation

The mean loss value for a population of IRs given random parameter values is around the 10–18 range (about 2.5–4.5 standard deviations per parameter) on the “Low” setting, with minimal improvements in fitness as the quality setting is increased. This indicates that not every combination of parameter values will be able to yield viable results. The relatively high error values for this first test can be partially attributed to the fact that certain combinations of parameter values would result in IRs that would be very difficult (or impossible) to produce. For instance, IRs with long $T_{60}$ tend to have lower clarity ($C_{80}$) values as well, since there are many more late reflections that can overpower the early ones. Otherwise, if reasonable values are chosen for these parameters, as in the second test, then IRs that closely match the desired parameter values can be obtained through the GA, with a mean error value of about 4 at “Low” quality and decreasing as quality increases.

### 5.2. Subjective Evaluation

As a whole, while the participants were able to distinguish between real and artificial IRs, two observations can be made: First, there is a significant decrease in the rate of correct responses when going from “High” quality to “Max” quality IRs in both the speech and music sections of the experiment. So even though the artificial IRs have yet to be indistinguishable from real IRs, the fact that such a drop occurred suggests that it is possible to minimize perceptual differences even further by increasing the execution time (e.g., by increasing the size of the population or the maximum number of generations). Second, the IRs are more distinguishable when convolved with music as opposed to speech. One possible explanation for this outcome is that the original speech and music signals were sampled at different frequencies (16 and 48 kHz, respectively). It is possible that the additional information (in frequencies above 8 kHz) could be used to identify differences in music and gain an advantage over speech. Furthermore, the impulsive nature of the drum riff could also make the differences more apparent (regardless of the difference in sample rate).

One question regarding these results would be: Given two different IRs with the same acoustic values, are they perceptually indistinguishable? According to Hak et al. [34], there is a Just Noticeable Difference (JND) for each of the ISO parameters in our solution: 5% for $T_{60}$, 5% for EDT, and 1 dB for $C_{80}$. Further experiments could determine the JND for combinations of these parameters and IRs in general.

### 5.3. Limitations

As previously mentioned, only monophonic IRs were evaluated. While binaural IRs can be produced by our solution, comparing them to real binaural IRs may hinder their perceptual evaluation, since spatial aspects, such as apparent location of the sound source (i.e., its azimuth and elevation), proximity, etc., may take relevance over the parameters we controlled (decay time, EDT, clarity, etc.).

As for the implementation of the plugin, while Matlab allows the prototyping and generation of VST plugins with relative ease, there are a few setbacks for a full implementation: There are a limited number of data types for parameters, meaning that some GUI elements such as buttons cannot be directly implemented. In practice, a toggle switch can act as a substitute for triggering actions such as generating an IR or saving a file, but the resulting interface may be clunky for an end user. One feature that could be improved in our solution is the capability to save the IRs on the disk. Matlab does not allow saving IRs directly as WAV files (or in most other audio formats), so binary files are used instead. These can be converted into WAV files using an ancillary program (an example of which is also implemented and provided in the plugin repository).

An additional area of improvement pertains to how the ILD can be controlled in our solution. Currently, a user can choose to either balance the stereo image (i.e., equalize the RMS levels of the IRs) or leave the ILD value to random chance. In the latter case, the ILD is clipped to ±20 dB only if it falls outside of this range. However, the ability to specify the ILD of the stereo image (e.g., from −20 dB to +20 dB) is another possibility. A revised implementation and subsequent change to the GUI is deferred to a future version. Another issue is the amount of CPU resources needed to both generate the impulse responses and process the input audio stream in real time, especially as the length of the IR increases. This is why the sample rate of the IRs in the plugin is set to 16 kHz and why the IR’s must then be resampled to match that of the input audio. Despite this, CPU overloading issues can still arise, especially when attempting to generate IRs with long $T_{60}$ times. Even with a CPU as powerful as the one used in this research, when running the plugin within Ableton Live 10 [35] at 44.1 kHz, the application often crashed when attempting to generate $T_{60}$ longer than about 2.5 s due to the long convolution. It is expected that, in the future, the full potential of multi-core processors becomes more accessible for real-time audio plugins so that either the partitioned convolution load can be distributed among several cores or the GA can be run as a background task while the convolution continues to run with the current IRs.

One more possible point of contention in our solution is the inclusion of the Bass Ratio parameter, since it only accounts for frequencies in the 125–2000 Hz range, meaning there is no control over frequencies outside of this range in the IR. Even Beranek [23] (p. 512) admits that BR is not very useful for measuring “warmth,” even though this method has been around for the longest. The decision to include the parameter was done for the sake of convenience to the user, who is thus able to partially control the frequency content of the reverberation with just a single parameter. One possible modification to our solution that could ameliorate this would be to allow the user to control the $T_{60}$, EDT, and $C_{80}$ values for individual octave bands, and to modify the fitness function of the GA to take these parameters into account for all octave bands. This would require adding additional parameters to the plugin’s GUI for each desired octave band, meaning that there is a trade-off between control over the artificial IRs and ease of use of the plugin.

## 6. Conclusions

A working prototype was developed for a plugin that uses GAs to create artificial reverberation. The parameters of the plugin have been carefully chosen and designed so that anyone from music producers to game designers could understand and utilize the plugin with ease. Even if current technologies prevent us from producing IRs with good enough fit within a reasonable amount of time, there is not necessarily a correlation between fitness value and desirability either. Perhaps the random nature of the GA can be seen as a feature rather than a setback to some users, as an IR that sounds pleasing to the ear may come at unexpected times. Thus, evolutionary algorithms can be a source of creativity in digital signal processing, audio design, and music production, as shown in our solution.

Further research and development could explore other methods that could decrease computation time and/or improve fitness, such as choosing different genetic operations or changing the genetic algorithm parameters. Parallelizing certain tasks with a multi-core CPU or a GPU, or at the very least executing the GA in the background, is one of several quality-of-life improvements that could also be made with further development. Whether or not it is possible to generate synthetic IRs that are perceptually indistinguishable from real-world IRs or other synthetic IRs with acoustic parameters within the JND margin is another topic for further investigation.

## Figures and Tables

**Figure 1 entropy-22-01309-f001:**
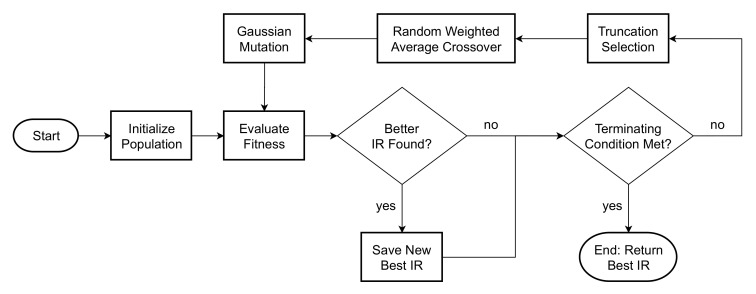
Flowchart outlining each step of the Genetic Algorithm (GA) used to generate an Impulse Response (IR).

**Figure 2 entropy-22-01309-f002:**
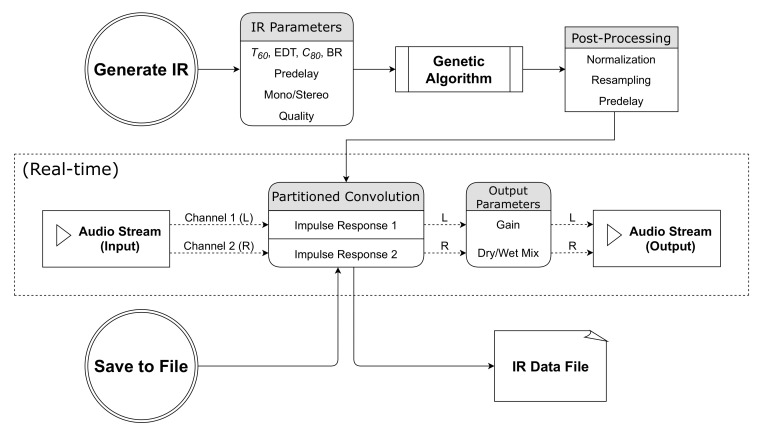
Flowchart illustrating the functionality of the plugin.

**Figure 3 entropy-22-01309-f003:**
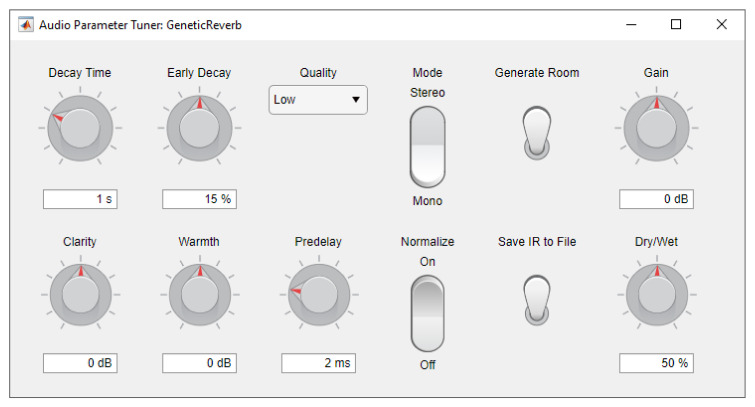
Program window displaying the plugin’s user interface.

**Figure 4 entropy-22-01309-f004:**
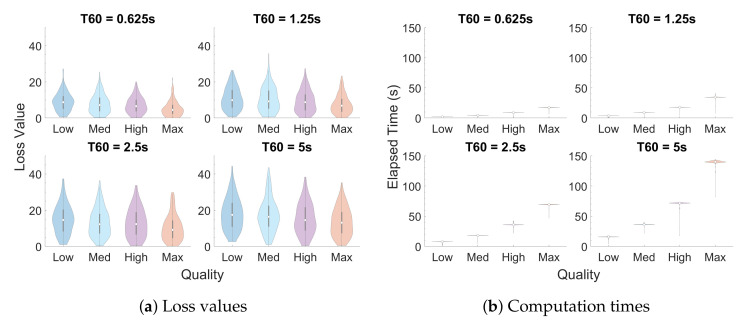
Violin plots of (**a**) loss values and (**b**) computation times for IRs generated with random acoustic parameter settings (lower is better). The center white dots and surrounding gray bars in these and other violin plots indicate the mean and interquartile range (IQR), respectively, of each distribution.

**Figure 5 entropy-22-01309-f005:**
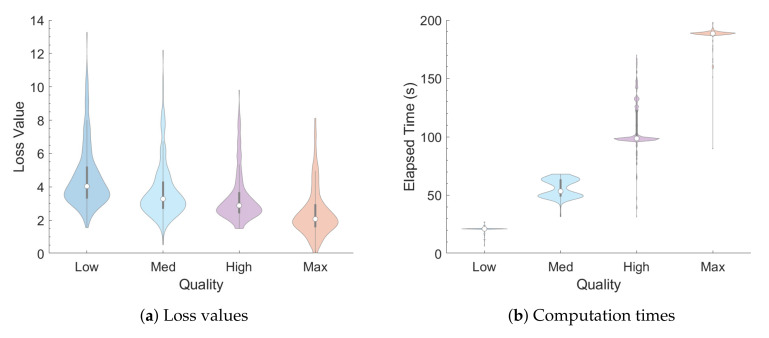
Violin plots of (**a**) loss values and (**b**) computation times for IRs generated with fixed settings (lower is better).

**Figure 6 entropy-22-01309-f006:**
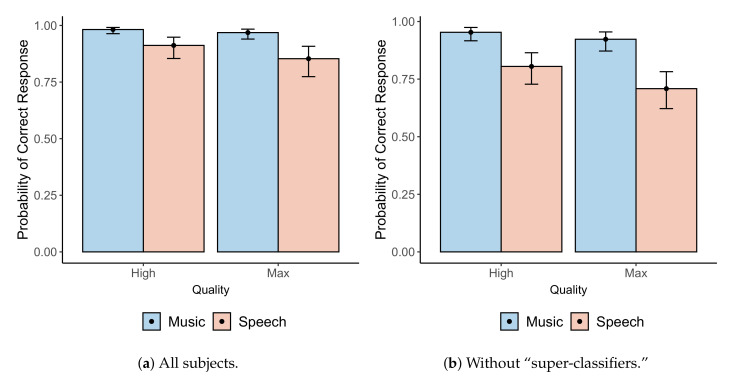
Means (dots) and 95% confidence intervals (error bars) for the rates of correct responses by quality setting and program: (**a**) for all subjects, and (**b**) after removing responses from subjects classified as “super-classifiers”.

**Table 1 entropy-22-01309-t001:** Acoustic parameters used in our solution, along with their colloquial names and definitions.

Parameter	Common Name	Definition
$T_{60}$	Decay Time, Reverberation Time	Time for sound pressure level of the IR to decay by 60 dB from the initial magnitude
Early Decay Time (EDT)	Reverberance	Time for sound pressure level of the IR to decay by 10 dB from the initial magnitude
$C_{80}$	Clarity	Ratio (in dB) between sound pressure levels of direct sound plus early reflections (<80 ms after direct sound) vs. late reflections (≥80 ms after direct sound)
Initial Time Delay Gap (ITDG)	Predelay, Intimacy	Time between the arrival of the direct sound and the arrival of the first reflection.
Bass Ratio (BR)	Warmth	Ratio between $T_{60}$ times in low-frequency (125–500 Hz) and mid-frequency (0.5–2.0 kHz) octave bands

**Table 2 entropy-22-01309-t002:** Parameter values used in the genetic algorithm within our solution depending on reverb quality. Selection rate, fitness threshold, and mutation probability were constant at 0.4, 0.1, and 0.001, respectively.

Parameter	Quality			
	Low	Med	High	Max
Population Size	25	25	50	50
Max. Number of Generations	20	50	50	100
Plateau Length	4	10	10	20

**Table 3 entropy-22-01309-t003:** IR-parameters used in the plugin with their valid ranges and descriptions.

Parameter	Valid Range	Description
Decay Time	[0.4,10] s	$T_{60}$ of the desired IR
Early Decay Time	[5,25] %	EDT of the desired IR (as a percentage of $T_{60}$)
Clarity	[−30,30] dB	$C_{80}$ of the desired IR
Warmth	[−10,10] dB	BR of the desired IR
Predelay	[0.5,200] ms	ITDG of the desired IR
Quality	{“Low,” “Medium,” “High,” “Max”}	Sets various parameters for the GA (see Table 2)
Mono/Stereo	{“Mono,” “Stereo”}	Generate either one IR for both channels (mono) or a different one for each channel (stereo)
Normalize	{“On,” “Off”}	In “stereo” mode, forces the RMS level difference in IRs to be zero (“On”) or at most 20 dB (“Off”)
Dry/Wet	[0,100]%	Balance between the dry input signal (0%) and the processed one (100%)
Output Gain	[−60,20] dB	Gain of the mixed dry/wet signal
Generate Room	N/A	Pressing this button generates a new IR using the current parameters
Toggle To Save	N/A	Pressing this button saves the current IR as a binary file in the plugin directory

**Table 4 entropy-22-01309-t004:** Chi-squared test statistics—including $\chi^{2}$ values, degrees of freedom (DF), and *p*-values— obtained from pairwise comparison of various Generalized Linear Mixed Models (GLMMs) defined by various formulas and fitted to experiment data. The terms hit ∼ (1|id) have been abbreviated to ∼. Significant *p*-values at a 95% confidence level are highlighted in bold.

Model 1	Model 2	$\chi^{2}$	DF	*p*
∼	∼ + (1|sex)	0.014	1	0.905
∼	∼ + (1|age)	0.008	1	0.928
∼	∼ + (1|exp)	0	1	0.994
∼	∼ + (1|rep)	0	1	0.999
∼	∼ + program	41.483	1	<0.001
∼ + program	∼ + program + quality	5.732	1	0.017
∼ + program + quality	∼ + program + quality + x.type	0.338	1	0.561
∼ + program + quality	∼ + program + quality + program:quality	0.191	1	0.662
∼ + program + quality	∼ + program + quality + program:x.type	0.848	2	0.654
∼ + program + quality	∼ + program + quality + quality:x.type	0.378	2	0.828
∼ + program + quality	∼ + program + quality + program:quality:x.type	3.671	5	0.598

**Table 5 entropy-22-01309-t005:** Chi-squared test statistics obtained from pairwise comparison of various GLMMs defined by various formulas and fitted to experiment data without super-classifiers. The terms hit ∼ (1|id) have been abbreviated to ∼. Significant *p*-values at a 95% confidence level are highlighted in bold.

Model 1	Model 2	$\chi^{2}$	DF	*p*
∼	∼ + (1|sex)	0	1	1.000
∼	∼ + (1|age)	0	1	1.000
∼	∼ + (1|exp)	0	1	1.000
∼	∼ + (1|rep)	0	1	1.000
∼	∼ + program	34.151	1	<0.001
∼ + program	∼ + program + quality	4.212	1	0.040
∼ + program + quality	∼ + program + quality + x.type	0.192	1	0.661
∼ + program + quality	∼ + program + quality + program:quality	0.326	1	0.568
∼ + program + quality	∼ + program + quality + program*x.type	0.527	2	0.769
∼ + program + quality	∼ + program + quality + quality*x.type	0.201	2	0.904
∼ + program + quality	∼ + program + quality + program*quality*x.type	3.535	5	0.618

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
