# Peer review of "Generating Artificial Reverberation via Genetic Algorithms for Real-Time Applications†"

_entropy, 2020, doi:10.3390/e22111309_

Round 1

Reviewer 1 Report

The authors present an interesting proposal for synthesizing artificial reverberant fields via genetic algorithms. In general, the article displays a suitable development that includes a well-based analysis of both objective and subjective results. Additionally adequate reflections are presents in the discussion section.

One relevant aspect is that the authors should provide clear clarifications in relation to the difference with the paper [Ly2020]. It is important to specify what is new in this article. It is also noteworthy that when subjecting the paper to Turniting software, a 21% similarity is obtained in relation to [Ly2020].

Some key points to improve the article are:

1. It is suitable to change the name to make it different from [Ly2020].
2. The introduction section needs to include the article’s contribution to demonstrate the difference with the article published in [Ly2020]; additionally, the authors must explicitly cite the article [Ly2020].
3. In the ‘‘Background’’ section add an introductory paragraph describing -briefly- the topics covered and how those are related to the paper.
4. The section 3 ‘‘Method’’ requires an introductory paragraph showing the purpose of that section.
5. It is suggested add an image in section 3.1, “Genetic Algorithm,” to graphically represent the genetic algorithm operation for the proposal made in the article.
6. Expand the introductory paragraph in section 4, “Evaluation” by defining the section purpose.
7. Expand the explanation of figures 3, 4, 5 and 6.
8. Expand the explanation or even the discussion of figure 7.
9. Add an introductory paragraph in section 5, “Discussion”.
10. Expand (enhance) the conclusions. Could you divide the conclusions into separated ideas?
11. Section 4 may have a justification for the type of test employed regarding the possibility of using parametric and non-parametric statistical tests.
12. A general revision of the article is suggested to improve the writing.

References:

[Ly2020] Edward Ly, Julián Villegas, Genetic Reverb: Synthesizing Artificial Reverberant Fields via Genetic Algorithms, Lecture Notes in Computer Science, vol. 12103. Springer, Cham. Artificial Intelligence in Music, Sound, Art and Design. EvoMUSART 2020.

Author Response

Thank you for your feedback. We apologize if the relation to our previous paper was not clear, but we hope this is now addressed in the new manuscript. Specifically, we have added a paragraph from l.42 in the introduction with a detailed relation of the changes we did from the previous manuscript.
Furthermore, some additional paragraphs, sentences, and a figure have also been added to address points 2-10.

As for the remaining points:

Point 1: The title of the paper is now "Generating artificial reverberation via genetic algorithms for real-time applications."

Point 11: Thank you for noticing this, regarding the selection of the statistical tool for the analysis of our data, we think that the use and benefits of GLMMs for binomially-distributed data has been discussed elsewhere. We included a sentence at the beginning of the "Results" section with an additional reference.

Point 12: We have made revisions throughout the manuscript for the purpose of improving clarity.

For full transparency, another change that we have made but was not addressed by the reviewers was the results of the subjective experiment. We discovered a flaw in our method when preparing the subjective experiment, namely that the acoustic parameter C80 was incorrectly calculated for both synthetic and real IRs. So we decided to fix this issue, run the experiment again, and update the manuscript with the new results. The main conclusion remains the same, but only a few minor details have been changed. Also, participants in this new experiment were financially compensated.

Reviewer 2 Report

This paper presents an approach to tuning gaussian-noise based IRs to match specified perceptual IR qualities via a genetic algorithm. The paper is well written and presented, and its great that the authors shared an implementation of the described method (although it's a shame that it's provided as MATLAB code rather than for an open language like Python or C++).

The explanation of the method, and especially the evaluation are of very good quality and should be commended. However, there's no real attempt to justify why this approach in particular is taken. Why are GAs a good fit to this problem as opposed to e.g. gradient descent? Why use Gaussian noise instead of other established reverb IR generation techniques like sparse/velvet noise? Why not use a constrained form of a standard algorithmic technique like an FDN? The paper would be much improved by addressing these issues, as otherwise the starting point feels rather arbitrary.

Specific comments:
L.205 Why would you want to replicate the background noise of an IR recording? This seems like an unusual choice, and therefore needs more justification.
L.221 This needs more justification or a reference.
L.303 It seems that generating the L/R IRs separately will likely result in an unbalanced stereo image. Did you consider either enforcing more similarity between the two sides, or perhaps generating Mid/Side IRs rather than L/R?

Author Response

Thank you for your feedback. We interleave your questions/comments (C) and our answers (A) below.

C: This paper presents an approach to tuning gaussian-noise based IRs to match specified perceptual IR qualities via a genetic algorithm. The paper is well written and presented, and its great that the authors shared an implementation of the described method (although it's a shame that it's provided as MATLAB code rather than for an open language like Python or C++).

A: We have considered porting our solution to other languages such as C++, but MATLAB was ultimately chosen for its ease of implementing and prototyping a VST software solution quickly, allowing us to focus on the methods rather than the implementation itself.

C: The explanation of the method, and especially the evaluation are of very good quality and should be commended. However, there's no real attempt to justify why this approach in particular is taken. Why are GAs a good fit to this problem as opposed to e.g. gradient descent?

A: It is true that many of the steps in our method are entirely arbitrary, and the steps we eventually chose in our method were the result of much trial-and-error. Given additional development time, it is possible that we may discover other methods that can produce better results, but the current method we produced so far is adequate enough for our purposes.

The question of why GAs are better for this problem was not properly addressed in the manuscript because we did not make a formal comparison with other methods. Rather, we specifically wanted to see if GAs were suitable for this kind of problem, and we found that, indeed, given some reasonable constraints GAs are suitable. We have added a sentence in the introduction to clarify our approach in response to this comment.

C: Why use Gaussian noise instead of other established reverb IR generation techniques like sparse/velvet noise? Why not use a constrained form of a standard algorithmic technique like an FDN?
The paper would be much improved by addressing these issues, as otherwise the starting point feels rather arbitrary.

A: We derived our method from that suggested in Zölzer (4th reference in the manuscript, p. 144) and added such reference at the beginning of section 3.1.1.

C: L.205 Why would you want to replicate the background noise of an IR recording? This seems like an unusual choice, and therefore needs more justification.
C: L.221 This needs more justification or a reference.

A: We wanted to try to replicate an IR recording as much as possible to minimize any perceptual differences between the synthetic and recorded IRs for the subjective evaluation. We have clarified our statements in these two lines and added references accordingly.

C: It seems that generating the L/R IRs separately will likely result in an unbalanced stereo image. Did you consider either enforcing more similarity between the two sides, or perhaps generating Mid/Side IRs rather than L/R?

A: Currently, we impose a limit on the level difference in cases where the stereo image is too unbalanced. Alternatively, a user can choose to completely balance the stereo image (i.e., equalize the IR levels). However, as the reviewer noted, we do not provide control on the lateralization of the resulting auditory image. An ILD controller (e.g., from -20 dB to +20 dB) is desirable and we defer it to a further version while we find the best way to reflect those changes in the GUI of the plugin. A note on this regard was added towards the end of the discussion.

For full transparency, another change that we have made but was not addressed by the reviewers was the results of the subjective experiment. We discovered a flaw in our method when preparing the subjective experiment, namely that the acoustic parameter C80 was incorrectly calculated for both synthetic and real IRs. So we decided to fix this issue, run the experiment again, and update the manuscript with the new results. The main conclusion remains the same, but only a few minor details have been changed. Also, participants in this new experiment were financially compensated.